# Molecular Epidemiology of *Candida Auris* Outbreak in a Major Secondary-Care Hospital in Kuwait

**DOI:** 10.3390/jof6040307

**Published:** 2020-11-21

**Authors:** Wadha Alfouzan, Suhail Ahmad, Rita Dhar, Mohammad Asadzadeh, Noura Almerdasi, Naglaa M. Abdo, Leena Joseph, Theun de Groot, Walid Q. Alali, Ziauddin Khan, Jacques F. Meis, Mohammad R. Al-Rashidi

**Affiliations:** 1Microbiology Unit, Department of Laboratories, Farwaniya Hospital, P.O. Box 13373, Farwaniya 81004, Kuwait; ritadhar50@hotmail.com (R.D.); n_almerdasi@me.com (N.A.); 2Department of Microbiology, Faculty of Medicine, Kuwait University, P.O. Box 24923, Safat 13110, Kuwait; suhail.ahmad@ku.edu.kw (S.A.); mohammad.assadzadeh@ku.edu.kw (M.A.); leena.anniejoseph@ku.edu.kw (L.J.); ziauddin381044@gmail.com (Z.K.); 3Department of Infection Control, Farwaniya Hospital, P.O. Box 13373, Farwaniya 81004, Kuwait; nagla_abdo@hotmail.com; 4Department of Community, Environmental and Occupational Medicine, Faculty of Medicine, Zagazig University, Zagazig 44519, Egypt; 5Department of Medical Microbiology and Infectious Diseases, Canisius Wilhelmina Hospital, 6532 SZ Nijmegen, The Netherlands; T.Groot@cwz.nl (T.d.G.); jacques.meis@gmail.com (J.F.M.); 6Department of Epidemiology and Biostatistics, Faculty of Public Health, Kuwait University, P.O. Box 24923, Safat 13110, Kuwait; w.alali@ku.edu.kw; 7Center of Expertise in Mycology, Radboud University Medical Center/Canisius Wilhelmina Hospital, 6532 SZ Nijmegen, The Netherlands; 8Bioprocess Engineering and Biotechnology Graduate Program, Federal University of Paraná, Curitiba 80060-000, Brazil; 9Hospital Administration, Farwaniya Hospital, P.O. Box 13373, Farwaniya 81004, Kuwait; alrashidi_mb@hotmail.com

**Keywords:** *C. auris*, outbreak, Farwaniya Hospital, candidemia, colonization, Kuwait

## Abstract

The emerging, often multidrug-resistant *Candida auris* is increasingly being associated with outbreaks in healthcare facilities. Here we describe the molecular epidemiology of a *C. auris* outbreak during 18 months, which started in 2018 in the high dependency unit (HDU) of a secondary-care hospital in Kuwait. Demographic and clinical data for candidemia and colonized patients were prospectively recorded. Clinical and environmental isolates were subjected to phenotypic and molecular identification; antifungal susceptibility testing by broth microdilution method; PCR-sequencing of *ERG11* and *FKS1* for resistance mechanisms to triazoles and echinocandins, respectively; and molecular fingerprinting by short tandem repeat (STR) analyses. Seventy-one (17 candidemic and 54 colonized) patients including 26 with candiduria and seven environmental samples yielded *C. auris*. All isolates were identified as *C. auris* by Vitek2, MALDI-TOF MS, PCR amplification and/or PCR-sequencing of rDNA. Twelve candidemia and 26 colonized patients were admitted or exposed to HDU. Following outbreak recognition, an intensive screening program was instituted for new patients. Despite treatment of all candidemia and 36 colonized patients, 9 of 17 candidemia and 27 of 54 colonized patients died with an overall crude mortality rate of ~50%. Nearly all isolates were resistant to fluconazole and contained the Y132F mutation in *ERG11* except one patient’s isolates, which were also distinct by STR typing. Only urine isolates from two patients developed echinocandin resistance with concomitant *FKS1* mutations. The transmission of *C. auris* in this outbreak was linked to infected/colonized patients and the hospital environment. However, despite continuous surveillance and enforcement of infection control measures, sporadic new cases continued to occur, challenging the containment efforts.

## 1. Introduction

*Candida auris* is an emerging pathogenic yeast that causes nosocomial invasive infections and outbreaks with high mortality rates, mostly in intensive care unit (ICU) patients [1,2,3,4]. It is now recognized as a formidable pathogen due to several unique characteristics which include its ability to persist on medical devices, hospital environment and provider hands allowing easy patient-to patient transmission despite absence of direct contact between them [5,6,7,8]. *C. auris* also has the ability to form biofilms and resist elimination by commonly used disinfectants and exhibits resistance to one or more commonly used antifungal drugs [5,6,7,8]. Although described as a novel bloodstream pathogen fairly recently [9], *C. auris* has caused invasive infections in many countries across all inhabited continents [1,10,11]. Once introduced into a setting/geographical location/country, *C. auris* spreads rapidly in health care settings [12,13,14]. Consequently, the epidemiology of invasive *Candida* infections has recently changed dramatically with *C. auris* becoming a major bloodstream pathogen in many health care facilities/geographical settings [14,15,16,17]. Although *C. auris* is highly clonal, whole genome sequence analyses have identified five distinct geographically restricted clades. These clades were initially detected among *C. auris* isolates from South Asia (clade I), East Asia (clade II), Africa (clade III), South America (clade IV) and Iran (clade V), which differ from each other by thousands of single-nucleotide polymorphisms [18,19,20]. Previous studies have also shown that *C. auris* isolates from Kuwait and other adjoining countries with a large expatriate population originating from South Asian countries belong to the South Asian clade (clade I) [14,21,22]. *C. auris* strains exhibit clade-specific resistance to fluconazole and varying susceptibility to other azoles, amphotericin B and echinocandins, often resulting in a multidrug-resistant phenotype for many isolates [11,14,19,23,24].

There is growing evidence that *C. auris* colonizes human body sites, particularly axilla, groin and nares and persists in hospital environments with the propensity for horizontal transmission in healthcare settings [3,5,6,7,8,25,26,27,28]. High mortality rates have been linked with invasive candidiasis caused by *C. auris* in susceptible patient populations, especially those with compromised immune system [5] and multiple comorbidities [2,3,6,29,30]. Although in the beginning only sporadic cases of invasive infections due to *C. auris* were detected soon after its recognition as a bloodstream pathogen, nosocomial outbreaks are now being reported with increasing frequency and involving larger patient cohorts [10,11,28,31]. *C. auris* mini/major outbreaks have been reported from the United Kingdom [32,33], United States of America [34,35], India [12,16,25], Spain [6,30,36], South Africa [13], Colombia [37], Kenya [38], Russia [39], Saudi Arabia [40] and Oman [21,41]. Here we describe molecular epidemiology of a nosocomial outbreak which started in the high dependency unit (HDU) of a secondary-care hospital (Farwaniya Hospital) in Kuwait with genetically identical clinical and environmental *C. auris* isolates.

## 2. Materials and Methods

### 2.1. Outbreak Setting, Case Definitions and Sample Collection

The outbreak likely began in January 2018 in the high dependency unit (HDU) of a major secondary-care hospital (Farwaniya Hospital with a capacity of 1200 beds) but was only recognized in September 2018 when three consecutive candidemia cases due to *C. auris* were detected within a short time span. The HDU with 20 beds is similar to an ICU but houses chronic patients as a long-term facility. However, patients are transferred to different hospital wards with improvement in their clinical condition. Thus, movement of patients to and from HDU and other hospital wards takes place depending on their medical requirements and condition. All patients that yielded *C. auris* from bloodstream or other anatomic sites from January 2018 to June 2019 were included in the study. The study was performed in accordance with Kuwait regulatory requirements and approval was obtained from the Ministry of health Ethical Committee, Kuwait (Approval number 1121/2019 dated 3 September 2019). The clinical specimens including blood were obtained from hospitalized patients after obtaining verbal consent only as part of routine patient care and diagnostic work-up for the isolation and antifungal susceptibility testing of fungal pathogens. All patient’s personal identifiers were kept confidential and the results are reported in this manuscript on deidentified samples without revealing patient identity.

Clinical cases were defined as symptomatic patients from whom a *C. auris* isolate was recovered from bloodstream or other sterile sites from January 2018 through June 2019. Invasive candidiasis was presumed when a patient showed raised inflammatory markers and β-D-glucan levels in serum despite treatment with broad-spectrum antibiotics and antifungal therapy. Persistent candidemia was considered when blood culture was positive for *C. auris* even after 7 days of antifungal treatment. Colonized cases were defined as hospitalized patients who did not have candidemia but yielded *C. auris* either from a clinical sample that was obtained for routine patient care (before identification of the outbreak, and these were retrospectively included) or from clinical samples from different anatomic sites specifically screened for *C. auris* colonization (prospectively included after identification of the outbreak in September 2018). The age variable among clinical and colonized cases was categorized into 4 age groups. Comorbidities were grouped into five main groups (i.e., hypertension, HTN; diabetes mellitus, DM; dyslipidemia, DLP; cardiovascular conditions, and “others”). The “others” group represented all comorbidities other than the four specified groups.

Depending on patient’s signs and symptoms, relevant clinical samples were collected for culture. Routine *C. auris* screening for all HDU patients was initiated from 19 December 2018. The patients were screened on the day of admission to HDU and then weekly until discharge. All hospitalized patients who had an epidemiologic link to a case patient or colonized patient in place or time were screened for the isolation of *C. auris*. Other patients that were apparently exposed to a *C. auris* case or the hospital environment where *C. auris* culture-positive patient was previously based were also screened. The surveillance samples were mostly obtained from axilla, groin, anterior nares, vascular line exit site, oropharynx, respiratory and/or urinary tract from patients with or without clinical signs of infection. Environmental samples processed in this study were obtained from rooms/units occupied by all *C. auris*-infected/colonized patients after the outbreak was recognized to determine contamination of the equipment/environment and the effectiveness of infection control and cleaning efforts. A total of 261 swab samples were processed for *C. auris* culture and were obtained from medical instruments, linen, walls/floor, furniture and high touch areas such as doorknob, bed railings/bedside drawer and toilet faucet/flush handles.

### 2.2. Laboratory Methods

All clinical samples collected from patients and environment were processed according to standard laboratory operational policy. The blood specimens were cultured in Bact T/Alert Blood Culture System (BD Diagnostics, Sparks, MD, USA) as described previously [42]. Surveillance samples from contacts were inoculated on blood agar with 5% sheep blood (BA) and Sabouraud dextrose agar (SDA), as described previously [43]. Environmental samples were inoculated on SDA containing 10% *w*/*v* sodium chloride plus 40 mg/L gentamicin. After incubation at 37 °C for 24–48 h, cultured colonies were Gram stained and yeast isolates were processed by Vitek2 automated yeast identification system (bioMerieux, Marcy I’Etoile, France) for provisional identification [42,43] with recently updated database. Species level identification of *Candida* isolates was performed by proteomic profiling by matrix-assisted laser desorption/ionization time-of-flight mass spectrometry (MALDI-TOF MS) (bioMerieux, France) [43], using updated reference database with *C. auris* spectrum included.

Definitive identification was performed by molecular methods. Genomic DNA from each isolate was prepared by the rapid boiling method using Chelex-100 as described previously [44]. Briefly, a loop full of *C. auris* colony grown on SDA plate was suspended in 1 mL of sterile water in a microcentrifuge tube containing 50 mg Chelex-100 (Sigma-Aldrich Co., St. Louis, MO, USA), the contents were heated at 95 °C for 20 min and then centrifuged. The supernatant was transferred to a new tube and was used as the source of genomic DNA. Typically, 2 μL was used for PCR amplification of various gene targets. *C. auris* identification was performed by PCR amplification of ITS region of rDNA by using *C. auris*-specific forward (CAURF, 5′-ATTTTGCATACACACTGATTTG-3′) and reverse (CAURR, 5′-CGTGCAAGCTGTAATTTTGTGA-3′) primers, as described previously [45]. Additionally, PCR amplification followed by DNA sequencing (PCR-sequencing) of the entire ITS region of rDNA was also performed for clinical and environmental isolates by using panfungal PCR amplification and sequencing primers, as described in detail previously [46]. Briefly, the ITS region of rDNA from each *C. auris* isolate was amplified by using panfungal ITS1 (5′-TCCGTAGGTGAACCTGCGG-3′) and CTS1R (5′-GCATATCAATAAGCGGAGGAAAAGA-3′) primers by following the PCR amplification protocol described previously [46]. The amplicons were purified by using PCR product purification kit (Qiagen, Hilden, Germany) used according to kit instructions. Both strands of purified amplicons were sequenced by using internal sequencing primers (ITS1FS, 5′-ACCTGCGGAAGGATCATT-3′; ITS2, 5′-TCGCTGCGTTCTTCATCGATGC-3′; ITS3, 5′-TCGCATCGATGAAGAACGCAGC-3′ and ITS4RS, 5′-GATATGCTTAAGTTCAGCG-3′) by following the sequencing protocol as described previously [46]. Isolates with sequence identity of >99% over the entire length of ITS region of rDNA with reference strains were confirmed as *C. auris* [2,14].

In vitro susceptibility to fluconazole (0.25 to 128 µg/mL), itraconazole (0.03 to 4 µg/mL), voriconazole (0.008 to 8 µg/mL), posoconazole (0.008 to 8 µg/mL), caspofungin (0.002 to 8 µg/mL), anidulafungin (0.002 to 8 µg/mL), micafungin (0.002 to 8 µg/mL) and amphotericin B (0.03 to 16 µg/mL) was determined by the broth microdilution procedure-based commercial colorimetric MICRONAUT-AM antifungal susceptibility testing (AST) panel for yeasts (Merlin Diagnostica GmbH, Bornheim, Germany) by following the manufacturer’s instructions. Quality control was ensured by simultaneously testing reference strains of *Candida krusei* (ATCC 6258) and *Candida parapsilosis* (ATCC 22019). The minimum inhibitory concentration (MIC) values were determined photometrically after 24 h of incubation at 35 °C as the drug concentration that inhibited 50% of the growth for all drugs except amphotericin B for which complete (>90%) growth inhibition was used and the data were interpreted according to European Committee on Antimicrobial Susceptibility Testing (EUCAST) method [47]. Susceptibility of environmental isolates was determined for fluconazole, voriconazole, amphotericin B and caspofungin by Etest, performed as described previously [2,45]. Although there are no established *C. auris*-specific susceptibility breakpoints, tentative susceptibility breakpoints of ≥32 µg/mL for fluconazole; ≥2 µg/mL for voriconazole, ≥2 µg/mL for posaconazole, ≥2 µg/mL for itraconazole, ≥4 µg/mL for anidulafungin, ≥4 µg/mL for micafungin, ≥4 µg/mL for caspofungin and ≥2 µg/mL for amphotericin B were used based on Clinical and Laboratory Standard Institute (CLSI) and EUCAST susceptibility data and expert opinion [19,23,48,49,50].

Common mutations conferring resistance to fluconazole in *C. auris* strains are located in hotspot I region of *ERG11* gene [23]. The *ERG11* mutations were detected by PCR amplification of *ERG11* by using *C. auris*-specific ERG11F (5′-GTGGGCTCTGCTGTTGTTTA-3′) and ERG11R (5′-CAAAACTTCCTCTTGGATTCTG-3′) primers and the PCR amplification protocol as described previously [14]. The amplicons were purified by using PCR product purification kit (Qiagen) used according to kit instructions. Both strands of purified amplicons were sequenced by using internal sequencing primers (ERG11FS, 5′-GCTCTGCTGTTGTTTACGGA-3′ or ERG11RS, 5′-ACTTCCTCTTGGATTCTGGGCA-3′) by following the sequencing protocol as described previously [14]. The mutations conferring resistance to micafungin in *C. auris* strains are located in hotspot 1 region of *FKS1* gene [23]. The *FKS1* mutations were detected by PCR amplification of hotspot 1 region of *FKS1* by using *C. auris*-specific FKS1F (5′-CTTTGGGTGGCTTGTTCACA-3′) and FKS1R (5′-CCAAGTAGAACGAACGACCA-3′) primers and the PCR amplification protocol as described previously [14]. The amplicons were purified by using PCR product purification kit (Qiagen) used according to kit instructions. Both strands of purified amplicons were sequenced by using internal sequencing primers (FKS1FS, 5′-TGGCTTGTTCACATCTTACA-3′ or FKS1RS, 5′-GTAGAACGAACGACCAATGGA-3′) by following the sequencing protocol as described previously [14].

Molecular fingerprinting of *C. auris* isolates was performed by 12-loci-based short tandem repeat (STR) typing, as described previously [51]. For this purpose, high quality *C. auris* DNA was extracted by using MagNA Pure 96 DNA and viral NASmall Volume Kit and MagNA Pure 96 instrument (Roche Diagnostics GmbH, Mannheim, Germany) by following the manufacturer’s instruction. Four multiplex reactions, which amplify 12 STR targets with a repeat size of 2, 3 or 9 nucleotides were performed for fingerprinting of the outbreak isolates [51]. The copy numbers of the 12 markers were determined using GeneMapper Software (Applied Biosystems). Relatedness between isolates was analyzed by using BioNumerics v.7.6.1 software (Applied Maths, Kortrijk, Belgium) by employing the unweighted pair group method with arithmetic mean averages (UPGMA) as described previously [52]. The data for outbreak isolates were also compared with environmental isolates collected from the same hospital, other *C. auris* isolates previously collected from Farwaniya and other hospitals in Kuwait and with *C. auris* isolates from other clades.

### 2.3. Infection Control Measures and Outbreak Management

Immediately after identifying the outbreak, an urgent meeting of the hospital infection control committee was held. The formal outbreak investigation led to the implementation of *C. auris* infection prevention and control measures according to the U.S. Centers of Disease Control and Prevention (CDC) guidelines, and Kuwait Infection Control Directorate (Ministry of Health) [53,54]. These measures included the following: (i) isolation of the infected/colonized patients under contact and standard isolation precautions in single occupancy room; (ii) strict adherence to hand hygiene; (iii) active surveillance screening from nose, axilla and groin for all patients on admission to the HDU and weekly thereafter for *C. auris* colonization; (iv) screening contacts of the newly identified colonized/infected patients; (v) cohorting of staff caring for *C. auris* infected/colonized patients; (vi) cleaning patient room environment with detergent and water followed by disinfection with 0.1% bleach (daily and final cleaning) according to the Pan American Health Organization/World Health Organization (PAHO/WHO) recommendations [27]; (vii) education and daily bed side training on *C. auris* infection control measures for all healthcare personnel; (viii) environmental screening, especially of high touch areas of rooms/facilities occupied by *C. auris* infected/colonized patients followed by thorough cleaning and disinfection of the affected rooms; (ix) proper cleaning and disinfection of all re-usable instruments and equipment; (x) waste and linen management according to the hospital policy; (xi) bundle of care to be applied for central venous catheter and urinary catheter insertion and maintenance; (xii) daily assessment of appropriateness of administration of antibiotics and antifungal agents by the clinical microbiologists and (xiii) minimizing the number of visitors to the unit. Active daily surveillance of infection in the HDU and all other hospital wards was performed by infection control team, in addition to follow up and monitoring the implementation and adherence to all the stipulated infection control measures.

### 2.4. Statistical Analysis

Descriptive statistics related to various patient data variables were provided. Categorical variables are presented as absolute number. The percentage of case-patients or colonized patients was compared across the levels of the study variables (i.e., gender, age-group, comorbidities, length of hospital stay before diagnosis [days], mortality and antifungal treatment) by using 2 × 2 chi-square or 2 × n likelihood ratio chi-square test, as appropriate. Statistical analyses were performed by using STATA software version 15.1 (Stata Corp., College Station, TX, USA). A *p* value <0.05 was considered as statistically significant.

## 3. Results

### 3.1. Patient Characteristics, Outbreak Description, Treatment and Outcome

The first documented *C. auris* isolate (Kw2027/17) from our hospital was obtained from the bloodstream of a candidemia patient in July 2017 [14]. Subsequently, bloodstream isolates from other patients were obtained in January, March and July 2018 and from the respiratory tract in March 2018 and from the urinary tract in May 2018 obtained from hospitalized patients during routine patient care. The outbreak was recognized when three consecutive bloodstream isolates were obtained within a short time period in September 2019. Altogether, 71 subjects were affected by the outbreak from January 2018 until June 2019 and included 17 candidemia patients (B1 to B17) with/without colonization with *C. auris* and 54 patients (C1 to C54) colonized with *C. auris* at one or more anatomic sites (Figure 1). Environmental screening of areas surrounding infected patients led to the isolation of seven *C. auris* isolates from surfaces such as bed rails, bed-side drawer, wall around the toilet, bathroom faucet handles and toilet flush handle.

Demographic characteristics of patients with invasive *C. auris* infection (candidemia) and those with mere colonization are presented in Table 1. There was no significant difference in the ratio of male to female among candidemia patients (males, 10 of 17, 58.8%) versus colonized patients (males, 37 of 54, 68.5%). Although candidemia patients were generally older; age distribution was nearly the same among candidemia and colonized patients, and the median patient age for candidemia cases and colonized patients was 71 and 59.5 years, respectively. Similarly, the length of hospital stay was usually longer for candidemia patients compared to colonized patients, and more colonized patients with <15 days of hospital stay were positive for *C. auris* than candidemia patients; however, the difference did not reach statistical significance (Table 1). Of 17 candidemia patients, 12 had been admitted to the HDU at least on one occasion during their hospital stay. Additionally, one patient was moved to the ICU while other patients were admitted in medical/surgical wards but were not directly exposed to either HDU or ICU during the outbreak. When all patients being admitted to the HDU were screened from December 2018, 3 of 10 (30%) of candidemia patients were found to be colonized in the axilla (*n* = 2), groin (*n* = 1) and nares (*n* = 1). Similarly, among 44 colonized patients that were actively screened from December 2018, 20 (45.4%), 20 (40.9%), 20 (40.9%), seven (15.9%), four (9.1%), two (4.5%) and one (2.3%) patient yielded *C. auris* from axilla, urine, groin, endotracheal tube, nares, central line and ear, respectively. Clinical diagnosis at the time of hospital admission in candidemia patients included pneumonia/bronchiectasis (*n* = 5), cerebrovascular accident (*n* = 3), septic shock (*n* = 2), cardiac arrest and pneumonia (*n* = 1), spinal abscess (*n* = 1), loss of consciousness and cerebrospinal fluid leak (*n* = 1), epilepsy (*n* = 1), pancytopenia (*n* = 1), chronic renal failure (*n* = 1) and intestinal obstruction (*n* = 1). Most patients also had multiple comorbidities like hypertension (*n* = 10), diabetes mellitus (*n* = 9), cardiovascular diseases (*n* = 7), dyslipidemia (*n* = 4) and others (such as hypothyroidism, pituitary macroadenoma, benign prostatic hyperplasia, chronic renal failure, bedridden status, retinitis pigmentosa, colon mass, obstructive lung disease and intracerebral hemorrhage) (*n* = 9) (Table 1). Multiple comorbidities were also present among 54 colonized patients and included hypertension (*n* = 30), diabetes mellitus (*n* = 30), cardiovascular diseases (*n* = 21), dyslipidemia (*n* = 8) and others (such as hypothyroidism, chronic/end-stage renal disease, ischemic heart disease, goiter and bedridden status) (*n* = 24). There was no significant difference in the comorbidities among candidemia patients versus colonized patients (Table 1).

All candidemia patients received antifungal treatment following identification of the yeast pathogen as *C. auris*. Treated patients either received caspofungin only (*n* = 12) or first caspofungin followed by amphotericin B (*n* = 2) or voriconazole only (*n* = 1) or first fluconazole followed by caspofungin (*n* = 1). Nine of 17 (52.9%) candidemia patients died including 5 of 10 patients whose repeat blood cultures were negative and 3 cases that had developed persistent candidemia despite therapy with caspofungin. Seven of 8 patients who survived had received caspofungin while one patient who was treated with voriconazole for 3 days was transferred to another facility. Although colonized patients remained blood culture-negative, 37 colonized patients received treatment with antifungal drugs empirically due to their critical condition. These included patients treated with caspofungin (*n* = 9), voriconazole followed by caspofungin (*n* = 5), fluconazole (*n* = 5), caspofungin followed by anidulafungin (*n* = 4), caspofungin followed by amphotericin B (*n* = 2), caspofungin followed by voriconazole (*n* = 2), fluconazole followed by amphotericin B (*n* = 2), anidulafungin (*n* = 2), amphotericin B (*n* = 2), amphotericin B followed by caspofungin (*n* = 1), voriconazole followed by fluconazole (*n* = 1), fluconazole followed by voriconazole (*n* = 1) and voriconazole (*n* = 1). The in-hospital mortality rate among colonized patients was nearly the same as in candidemia patients since 27 of 54 (50%) colonized patients died including 21 patients who received antifungal treatment (Table 1).

### 3.2. Culture Identification and Genotyping

Multiple isolates were cultured from many patients. Body sites more frequently yielding *C. auris* included urinary tract (*n* = 50), bloodstream (*n* = 23, including repeat cultures), axilla (*n* = 35), groin (*n* = 25), trachea (*n* = 8) and nares (*n* = 8). Blood culture was repeated in 13 of 17 patients 2–10 days after initiation of antifungal therapy and was negative in 10 of 13 patients. The culture of central venous catheter tip grew *C. auris* in two patients without corresponding candidemia. Of 261 swab samples taken during environmental screening, seven samples were positive for *C. auris*. Two of seven culture-positive swab samples (one each from bedside drawer and bedrail) were taken from an isolation room while the remaining five positive samples (one each from bedrail, bedside drawer, toilet flash handle, toilet faucet handle, wall around the toilet) were taken from a private room housing a heavily colonized patient. All clinical and environmental isolates were identified as *C. auris* by phenotypic (Vitek2 yeast identification system with updated database) method as well as by MALDI-TOF MS and PCR amplification of ~276 bp amplicon derived from the ITS region of rDNA [44]. Furthermore, the DNA sequences of the ITS region of rDNA for all the clinical and environmental isolates were identical and matched completely with the corresponding sequences of *C. auris* strains described previously from Kuwait and also with sequences of reference and other strains from India (South Asian clade) but were different from the sequences of *C. auris* isolates from Japan/Korea, South Africa, Venezuela and Iran.

Molecular fingerprinting performed by STR typing showed that nearly all outbreak isolates from both candidemia and colonized patients as well as the environmental isolates were genetically identical. Interestingly, the same STR typing pattern was also shared by the first documented *C. auris* isolate (Kw2027/17) obtained from a candidemia patient from Farwaniya Hospital in 2017. Only four isolates from one colonized patient (C5) differed at a single locus (Figure 2). Other *C. auris* strains previously isolated from other hospitals in Kuwait as well as *C. auris* isolates belonging to South Asian clade I described from India and Oman were also very closely related with our outbreak isolates as they also differed at one locus only (Figure 2).

### 3.3. Antifungal Susceptibility Testing and Mutation Analyses of ERG11 and FKS1 Genes

The AST results for 16 bloodstream isolates from 15 candidemia patients are presented in Table 2. A total of 15 isolates were resistant to fluconazole and voriconazole while one isolate (Kw2861/18) exhibited reduced susceptibility (MIC of 16 µg/mL) to fluconazole but was susceptible (MIC of 0.031 µg/mL) to voriconazole. Two isolates (including isolate Kw2861/18) were susceptible while the remaining 14 isolates were resistant to itraconazole. However, the isolates exhibited variable susceptibility to posaconazole (Table 2). All bloodstream isolates were susceptible to anidulafungin and micafungin, however, they exhibited variable susceptibility (MIC of 0.031 to 8 µg/mL) to caspofungin. Although all bloodstream isolates were also susceptible to amphotericin B, eight isolates exhibited reduced susceptibility (MIC of 1 µg/mL) to this drug. The AST results for 46 *C. auris* isolates from 30 colonized patients are presented in Table 3. Forty of 46 (87%) isolates were resistant while the remaining six isolates exhibited reduced susceptibility (MICs of 8 µg/mL and 16 µg/mL) to fluconazole. However, only 13 and 18 isolates were resistant to voriconazole and itraconazole, respectively (Table 3). Forty-three of 46 isolates were susceptible; one isolate from a colonized (C18) patient exhibited reduced susceptibility (MIC of 0.5 µg/mL), and two isolates from another colonized (C5) patient were resistant to anidulafungin and micafungin (Table 3). Like bloodstream isolates, caspofungin MIC values again varied considerably (0.25 to 8 µg/mL) among colonizing strains. All colonizing strains were susceptible; however, 18 isolates exhibited reduced susceptibility (MIC of 1 µg/mL) to amphotericin B (Table 3). By Etest, all environmental samples were uniformly resistant to fluconazole but susceptible to voriconazole, amphotericin B and caspofungin.

*C. auris* isolates obtained from all 17 candidemia patients, 53 colonized patients and all environmental samples contained Y132F mutation in *ERG11*. However, four isolates from one colonized patient (C5) contained K143R mutation. Similarly, *C. auris* isolates from all 17 candidemia patients, 52 colonized patients and all environmental samples contained wild-type sequence of hotspot-1 of *FKS1*. Although the initial urine isolates from the remaining two colonized patients (C5 and C18) contained wild-type sequence of hotspot-1 of *FKS1,* subsequent isolates either contained a nonsynonymous (S639F) mutation (from C5 patient) or a deletion (Δ635F) mutation (from C18 patient). Patient C5 was treated with caspofungin followed by voriconazole while patient C18 received treatment with anidulafungin and caspofungin and the three isolates with *FKS1* mutations were obtained after treatment with echinocandins. Interestingly, colonizing strains from other anatomic sites (axilla, groin or nares) from C18 patient contained wild-type sequence of hotspot-1 of *FKS1* while colonizing strains from other anatomic sites from C5 patient were not available.

### 3.4. Infection Control Measures and Outbreak Management

Immediately after identifying the outbreak, an urgent meeting of the hospital infection control committee was held. The formal outbreak investigation led to the implementation of *C. auris* infection prevention and control measures as described under “Materials and Methods”. Implementation of improved and robust infection control measures resulted in no new case of invasive infection during June to November 2019 and the number of colonized patients also declined sharply in the HDU; however, sporadic cases continued to occur in other wards. New cases started to appear again in February 2020, but the screening of patients was discontinued from March 2020 due to emergence of COVID-19 and re-allocation of available staff and resources towards this pandemic.

## 4. Discussion

The first case of invasive *C. auris* infection in Kuwait, an Arabian Gulf country in the Middle East, was detected in 2014 [55]. Subsequently, *C. auris* spread rapidly across various health care facilities within the country, and it became a significant bloodstream fungal pathogen in 2018 [14,45]. Here, we describe a *C. auris* outbreak in Kuwait which started in the HDU of a secondary-care facility (Farwaniya Hospital) and involved 17 candidemia patients and 54 colonized patients. Interestingly, *C. auris* was also isolated from the ear canal of one patient, an anatomic niche which had not yielded this organism in our previous studies [14,45]. Environmental screening of patient’s rooms and surroundings also yielded genetically identical *C. auris* isolates supporting previous observations that positive patients shed *C. auris* into the environment which can survive on both moist and dry surfaces for several days and pose a risk for continuous transmission to susceptible patients [3,6,8,21,30,32]. *C. auris* also persists on reusable skin-surface axillary temperature probes [8], which also supports higher likelihood of axilla being colonized more often than other body sites. Although we observed that many (20 of 44, 45.4%) patients were colonized with *C. auris* in the axilla, colonization of urinary tract (18 of 44, 40.9%) and groin (18 of 44, 40.9%) was also common among 44 colonized patients, who were actively screened since December, 2018. However, none of the 54 colonized patients developed candidemia, a finding similar to data reported previously from Spain [6].

Fifteen of 16 isolates from candidemia and 40 of 46 isolates from colonized patients were resistant while the remaining isolates exhibited reduced susceptibility to fluconazole at 24 h reading. Higher overall percentages of essential agreement between Etest and broth macrodilution MICs have been reported for fluconazole at 24 h compared to 48 h incubation time for clinical isolates of *C. albicans, C. glabrata* and *C. tropicalis* but not for *C. parapsilosis* [56]. Nearly all bloodstream isolates were also resistant to voriconazole and/or itraconazole while cross-resistance to these triazoles was lower among colonizing strains. Cross-resistance between fluconazole and other triazoles among non-*albicans Candida* spp. is not always complete [57]. Four (all bloodstream) of 11 fluconazole-resistant *C. parapsilosis* isolates and the two *C. lusitaniae* isolates with reduced susceptibility to fluconazole in Kuwait were also scored as susceptible to voriconazole [58,59]. The outbreak was confirmed by complete molecular identity of nearly all clinical and all environmental *C. auris* isolates analyzed in this study as they shared identical ITS region of rDNA sequences and identical STR typing patterns. The STR typing pattern of the outbreak isolates was also shared with the first documented *C. auris* isolate (Kw2027/17) obtained from a candidemia patient from our hospital in 2017 [14]. Only four isolates from one colonized patient (C5) showed a slightly different STR typing pattern that differed only in one (M3-1) STR marker. This pattern was exhibited by few *C. auris* isolates from another (Al-Sabah) hospital in Kuwait. Although patients are frequently transferred among various hospitals within Kuwait, apparently this (C5) 72-year-old patient was admitted in our hospital as a new patient from the community, and the first *C. auris* isolate was cultured from endotracheal tube nearly 20 days later. The first urine isolate was grown nearly a month after admission. Other *C. auris* isolates obtained several years apart from different hospitals (including other isolates from Al-Sabah Hospital) in Kuwait [14,45] were also closely related to outbreak isolates as they also differed in only one (M3-1) STR marker and were identical to clade I isolates from India and Oman [21]. The STR data also showed that there is a protracted continuous transmission of *C. auris* clade I isolates over the years throughout Kuwait which have undergone microevolution in different hospitals resulting in three distinct genotypes (denoted as genotype a, b and c) in clade I; genotype c likely evolved in Farwaniya Hospital and caused the outbreak described here.

All four isolates from the C5 colonized patient, obtained from urine samples, contained K143R mutation and were resistant to fluconazole but were susceptible to other azoles. All other clinical and environmental isolates from this outbreak contained Y132F mutation in *ERG11* and were either resistant or exhibited reduced susceptibility to fluconazole. Healey et al. [60] also reported that clade I and clade IV isolates with Y132F mutation exhibit cross-resistance to voriconazole while isolates with K143R mutation remain susceptible to voriconazole. However other studies with clade I isolates have reported variable results [10,23]. It is pertinent to mention here that the K143R mutation in *ERG11* was not detected previously from this hospital but was detected mainly among *C. auris* isolates from two other major hospitals (Al-Sabah and Ibn-Sina) located nearly 20 km from this facility [14].

*C. auris* has previously been isolated with increasing frequency from Kuwait [14,45] and other Middle Eastern countries [22], and outbreaks have also been reported recently from two other (Oman and Saudi Arabia) nearby countries [21,40,41]. These reports together with our study and various outbreaks reported recently from other geographical locations [3,6,21,30,33,36,37,38,39,40] strongly support the emerging role of *C. auris* as a major cause of invasive infections and outbreaks in health care facilities worldwide that have been difficult to control. *C. auris* has presumably assumed this role due to several unique characteristics. It exhibits resistance to many commonly used antifungal agents and has unique ability to persist and remain viable for months, likely due to formation of “dry” biofilms on hospital surfaces and environment [8]. It resists removal despite vigorous cleaning of patient’s room surfaces and environment, clothing and equipment with common disinfectants and exhibits higher propensity for easy transmission in hospital environment [5,23,30,47,53,61]. Furthermore, aggregative phenotypes of *C. auris* that are predominantly isolated from colonized patients have a higher capacity for biofilm formation than non-aggregative phenotypes and are not easily eradicated [62,63]. Consistent with these characteristics of *C. auris*, the outbreak in our hospital, which was initially contained due to implementation of improved and robust infection control measures, was not completely controlled as sporadic cases continued to occur in other wards and new cases started to appear again in HDU in February 2020.

The AST data showed that three *C. auris* isolates from two colonized (C5 and C18) patients were either resistant or exhibited reduced susceptibility to anidulafungin/micafungin while the remaining isolates were susceptible. The AST data for caspofungin were highly variable which are consistent with previous reports showing paradoxical growth (also known as Eagle effect) of *C. auris* isolates in the presence of caspofungin [33,64]. All three isolates with reduced susceptibility to echinocandins were obtained from urine samples. The two isolates from C5 patient were highly resistant (MIC of 8 µg/mL) to micafungin and contained S639F mutation while one isolate from C18 patient exhibited reduced susceptibility (MIC of 0.5 µg/mL) to micafungin and contained ΔF635 (deletion of codon 635) mutation in *FKS1*. Interestingly, both C5 and C18 colonized patients received echinocandins before isolates with reduced susceptibility were cultured. Other urine isolates obtained before treatment from C5 and C18 patients were susceptible to echinocandins and exhibited wild-type sequence of *FKS1*. The S639F is a common mutation detected in hotspot 1 region of *FKS1* in echinocandin-resistant *C. auris* isolates described in other studies [14,23,33,64] while ΔF635 mutation has also been described in in vitro-generated echinocandin-resistant *C. auris* and also among clinical echinocandin-resistant *C. glabrata* [65,66]. Although candidemia was not confirmed by culture in C5 and C18 patients, one previous study showed development of echinocandin resistance in a patient with recurrent *C. auris* candidemia secondary to chronic candiduria and the bloodstream isolate contained S639P mutation in hotspot 1 region of *FKS1* [67]. It seems that urinary tract and other non-sterile anatomic sites are favorable niches for the development of resistance in *Candida* spp. for echinocandins and amphotericin B [14,66,68,69,70].

Only 17 of 71 (24%) patients affected by the *C. auris* outbreak in our hospital developed candidemia. The data analyses from previous *C. auris* outbreaks have also shown that only a fraction (~14% to 40%) of affected patients develop candidemia [21,30,32,36,40]. The risk factors for invasive *C. auris* infections are similar to those for other *Candida* spp. [6,19,30,32]. These include multiple underlying medical conditions such as diabetes mellitus, malignancy, chronic kidney disease, neutropenia, prior or concomitant bacterial infection, use of broad-spectrum antibiotics and antifungal agents, recent surgery, presence of central venous catheters or urinary catheters, stay in ICU and total parenteral nutrition [2,19,25,32]. The most common comorbidities/underlying conditions identified among *C. auris*-infected patients in a meta-analysis carried out in 2018 included diabetes mellitus (>52%), sepsis (>48%), lung disease (>39%) and kidney disease (32%) [29]. More recently, Ruiz-Gaitan et al. [6] reported polytrauma, cardiovascular disease and cancer while Al-Maani et al. [21] reported diabetes mellitus, hypertension and cardiovascular disease as the major comorbidities. Khan et al. [2] reported prior use of broad-spectrum antibiotics, placement of central venous catheter, cancer/lymphoma and gastrointestinal abnormality/surgery as major risk factors for developing *C. auris* fungemia. Consistent with these observations, most of our candidemia and colonized patients also presented with several comorbidities which mainly included hypertension, diabetes mellitus, cardiovascular diseases, dyslipidemia and others (such as hypothyroidism, chronic/end-stage renal disease).

Although all candidemia patients and 37 of 54 colonized patients (due to their serious condition as a result of several comorbidities and underlying conditions) received antifungal treatment, a crude mortality rate of ~50% was observed among both candidemia and colonized patients during the outbreak in our hospital. However, none of the deaths were directly attributable to *C. auris* infection, and use of echinocandins did not reduce skin colonization. Highly variable crude mortality rates, varying from 0% to 72%, have been observed among *C. auris*-infected patients, likely reflecting the highly variable patient populations in different settings with different degrees of comorbidities, particularly for ICU patients and the susceptibility patterns of infecting strains [6,19,21,29,30,31,32,36,40,41].

In conclusion, 71 (17 candidemic and 54 colonized) patients were affected during the first 18 months of the ongoing *C. auris* outbreak in Farwaniya Hospital in Kuwait. All clinical and environmental isolates were identified as *C. auris* by phenotypic and molecular methods and, excluding four isolates from one patient, were genotypically identical. Nearly all isolates were resistant to fluconazole, and many isolates were also resistant to other triazoles. Only urine isolates from two patients, treated with caspofungin and/or anidulafungin, exhibited reduced susceptibility to echinocandins with concomitant *FKS1* mutations. Despite treatment of all candidemia and 36 colonized patients, an overall crude, in-hospital mortality rate of ~50% was seen among both candidemia and colonized patients. Our inability to decolonize patients due to unique properties of *C. auris* to cause prolonged host colonization contributed to the sustenance of the organism causing invasive infections at locations other than the HDU. Despite continuous surveillance and enforcement of infection control measures, new cases continued to occur, challenging the containment efforts.

The DNA sequencing data reported in this study have been submitted to GenBank under accession no. MW078445 to MW078481 and MW088578 to MW088709.

## Figures and Tables

**Figure 1 jof-06-00307-f001:**
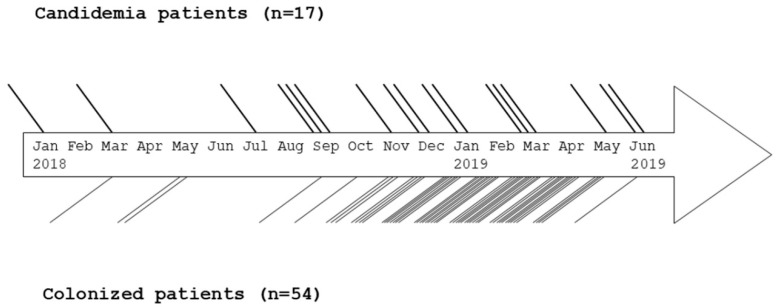
Timeline for the detection of patients with candidemia and colonization due to *C. auris* in Farwaniya Hospital during the course of outbreak from January 2018 to June 2019. Number of thick and thin lines indicate number of candidemia and colonized patients, respectively, detected in each month and the spacing between the lines within a month or between months is not relevant. Some candidemia patients also yielded *C. auris* from other anatomic sites, but they are not included among colonized cases. Multiple *C. auris* isolates were obtained over time from many anatomic sites from most of the colonized patients. None of the colonized patient yielded *C. auris* from bloodstream.

**Figure 2 jof-06-00307-f002:**
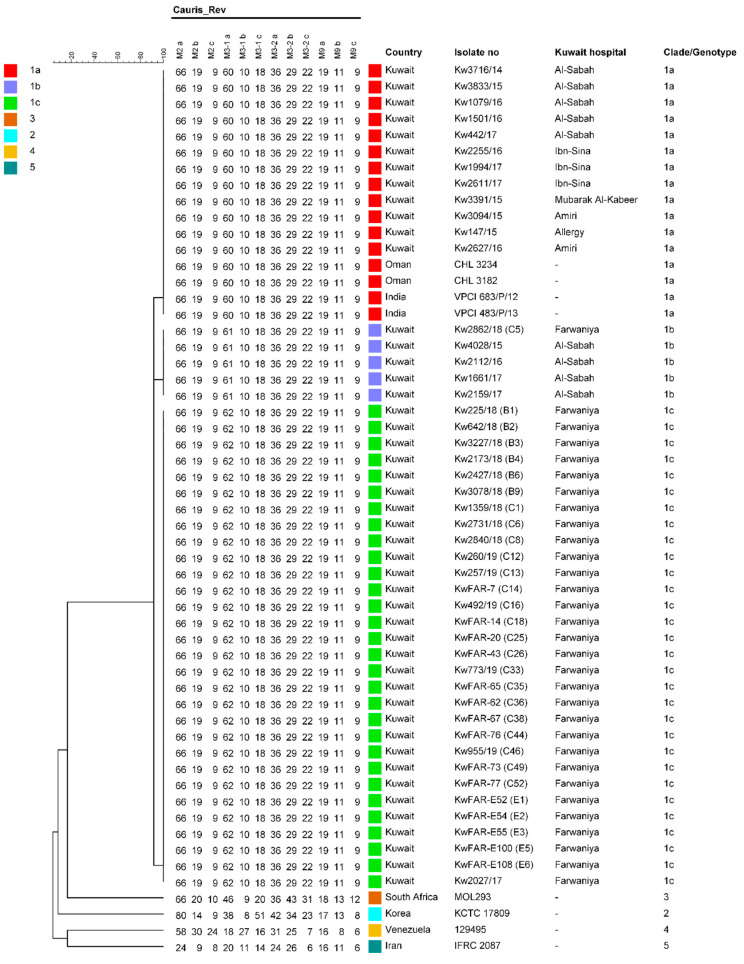
Short tandem repeat (STR) typing patterns of *C. auris* outbreak isolates from Farwaniya Hospital in Kuwait. Only one isolate from each patient is shown. Multiple isolates from each patient from same or different sites exhibited identical patterns. The first *C. auris* isolate (Kw2027/17) obtained from this hospital in 2017 as well as isolates from other hospitals in Kuwait are also included for comparison purpose. Representative South Asian clade 1 isolates from India and Oman and clade 2 to 5 isolates from Korea, South Africa, Venezuela and Iran, respectively, are also shown. The source of outbreak isolates and patient no. are also shown. (B1) = bloodstream isolate from candidemia patient 1; (C1) = colonizing strain from colonized patient 1; (E1) = environmental isolate 1. The scale in the upper left corner represents similarity (%).

**Table 1 jof-06-00307-t001:** Demographic and other characteristics of candidemia patients versus patients colonized with *C. auris.*

Variables	Candidemia Patients	Colonized Patients	*p* Value
	*n* = 17 (%)	*n* = 54 (%)	
Gender			
Male	10 (58.8)	37 (68.5)	0.461
Female	7 (41.2)	17 (31.5)	
Age (years) groups			
<50	3 (17.6)	16 (29.6)	0.381
50–65	3 (17.6)	15 (27.8)	
66–80	8 (47.1)	19 (35.2)	
>80	3 (17.6)	4 (7.4)	
Comorbidities			
Hypertension	10 (58.8)	30 (55.6)	0.813
Diabetes mellitus	9 (52.9)	30 (55.6)	0.85
Dyslipidemia	4 (23.5)	8 (14.3)	0.403
Cardiovascular diseases	7 (41.2)	21 (38.9)	0.866
Others ^a^	9 (52.9)	24 (44.4)	0.54
Length of hospital stay ^b^ (days)			
<15	1 (5.9)	16 (29.3)	0.068
15–35	5 (29.4)	17 (32.5)	
36–55	3 (17.6)	11 (20.4)	
>55	8 (47.1)	26 (48.2)	
Treatment with antifungals			
None	0 (0.0)	17 (31.4)	0.002
One antifungal drug	14 (82.4)	19 (35.2)	
Two antifungal drugs	3 (17.6)	18 (33.3)	
Mortality			
Yes	10 (58.8)	27 (50)	0.44
No	7 (41.2)	27 (50)	

^a^ Other comorbidities included hypothyroidism, pituitary macroadenoma, benign prostatic hyperplasia, chronic renal failure, bedridden status, retinitis pigmentosa, ischemic heart disease, colon mass, obstructive lung disease, intracerebral hemorrhage etc. ^b^ Before detection of *C. auris* in any clinical sample.

**Table 2 jof-06-00307-t002:** In vitro antifungal susceptibility testing results of *C. auris* bloodstream isolates (*n* = 16) against fluconazole, voriconazole, itraconazole, posaconazole, anidulafungin, micafungin, caspofungin and amphotericin B after 24 h of incubation at 35 °C.

Antifungal	Number of Isolates with Minimum Inhibitory Concentration (MIC) Value (µg/mL) of																	Geometric
Drug	0.002	0.004	0.008	0.016	0.031	0.062	0.125	0.25	0.5	1	2	4	8	16	32	64	128	Mean (µg/mL)
Fluconazole														1 *	**1**	**12**	**2**	61.29
Voriconazole					1							**15**						2.95
Itraconazole					1				1			**14**						2.59
Posaconazole			1		2			1	4	7 *	**1**							0.39
Anidulafungin				10	6													0.02
Micafungin				12	4													0.02
Caspofungin					1			1		2 *	**3**	**5**	**4**					2.18
Amphotericin B									8	8 *								0.71

Isolates with MICs indicative of resistance to antifungal drugs are highlighted in bold numbers. Isolates with MICs indicative of reduced susceptibility to antifungal drugs are indicated by asterisk (*). The modal values are underlined.

**Table 3 jof-06-00307-t003:** In vitro antifungal susceptibility testing results of *C. auris* isolates (*n* = 46) from colonized patients against fluconazole, voriconazole, itraconazole, posaconazole, anidulafungin, micafungin, caspofungin and amphotericin B after 24 h of incubation at 35 °C.

Antifungal	Number of Isolates with Minimum Inhibitory Concentration (MIC) Value (µg/mL) of																	Geometric
Drug	0.002	0.004	0.008	0.016	0.031	0.06	0.125	0.25	0.5	1	2	4	8	16	32	64	>128	Mean (µg/mL)
Fluconazole													3 *	3 *	**4**	**31**	**5**	51.83
Voriconazole				1	8	9	5	6	4		**1**	**12**						0.27
Itraconazole					12	10	1	1	3	1 *	**1**	**17**						0.36
Posaconazole			27	4	6	1	1		2	4 *			**1**					0.02
Anidulafungin				26	13		3	1	1 *			**1**	**1**					0.03
Micafungin				37	6				1 *				**2**					0.02
Caspofungin								12	10	4 *	**2**	**9**	**9**					1.22
Amphotericin B									28	18 *								0.66

Isolates with MICs indicative of resistance to antifungal drugs are highlighted in bold numbers. Isolates with MICs indicative of reduced susceptibility to antifungal drugs are indicated by asterisk (*). The modal values are underlined.

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
