# Peer review of "Molecular Epidemiology of Candida Auris Outbreak in a Major Secondary-Care Hospital in Kuwait"

_jof, 2020, doi:10.3390/jof6040307_

Round 1
Reviewer 1 Report
This is an overview of an investigation of an ongoing C. auris outbreak in an acute care hospital in Kuwait. The authors provide a well-rounded description of the patients and human and environmental samples from epidemiologic and laboratory perspectives. Their comprehensive approach to the investigation and subsequent description touches on several issues of interest in the field, including body site positivity, candidemia v. colonization or non-invasive infections, environmental contamination, patient outcomes, drug resistance, genetic relatedness, and, however briefly, how COVID-19 is affecting C. auris response.
Line 51-52: The authors should clarify this part of the sentence: “…its capability to evade immune response and easy person-to-person transmission through direct/indirect contact…” As it is, it reads to be like it is difficult to transmit and also that that paradoxically makes it a “formidable foe.” I think perhaps the intent is more that it is very easily transmitted by medical equipment and provider hands and thus still transmits easily despite the patients not being in direct contact with each other.
Line 59: I am not sure “originating” is appropriate here as it might be subject to finder’s bias. We do not necessarily know that those clades originated in those regions.
Line 93-94: This is suggesting that there were no cases that where asymptomatic and the authors happened to find C. auris rather than collecting samples to specifically look for this. However, this is common, so the authors may want to clarify whether this did not happen or whether such patients were counted as having clinical or colonized cases.
Line 93-94: There was a timeframe provided for the clinical cases, but not for the colonized cases. This should be added if available. For instance, are the authors only counting the point from which all patients were screened onward or did they begin screening select patients prior to that?
Line 97: “Contacts” is defined here, but not used elsewhere in this paragraph to understand what happened with contacts. If they were saying that contacts were screened, I would recommend being more explicit or using the term “contact” elsewhere in that paragraph.
Line 102: This line says environmental samples were obtained from “rooms/facilities,” but I thought that all sampling in this study was from the same facility. Should this be buildings or units instead of facilities?
Lines 103-105: Check plurality and punctuation in this sentence.
Line 139: The citation here is 47, but the EUCAST citation in the references is 48. The authors must look through this document and make sure all citations are correctly lining up with their intended references. It looks like this is perhaps caused by the title “References” showing up at #1 in the reference list.
Line 210: Change isolated to isolates.
Figure 1: It is not clear how colonized cases were identified prior to the identification of the outbreak (i.e. prior to September 2018). Did those patients have unusual circumstances that caused the authors to want to screen them even though they were unaware of C. auris within the facility (e.g. healthcare exposure elsewhere in the country or abroad known to have C. auris)? This makes me continue to question the clinical v. colonized case definition, as asked about in my Line 93-94 question above.
Figure 1: It might help to have more of a standard format for the epi curve. Here, it looks like there is a break between each month where no cases were found, but I am guessing it was more continuous than an oscillating up and down curve and these breaks are rather just a result of how to graphic is made.
Figure 1: It would be helpful to clarify if any patients who were first found to have C. auris due to a screening but then they subsequently developed C. auris candidemia. And, if so, how are these categorized? I’m wondering if that distinction would change the timeline for these cases at all.
Line 227: Please check to see if IQR was meant instead of 95% CI in the parentheses.
Line 234-237: I would recommend putting the percentages next to the body site given the list is so long and indicating what the denominators were. Also, it is not clear if the authors are saying that the patients with candidemia were screened after they were known to have candidemia already or if patients who were screened and then went on to have candidemia. From 358-359 in the discussion makes it sound like the former, but this would be helpful to clarify earlier on. Also, were the patients with candidemia not screened at the other body sites or were they negative? Who were the “44 other patients”? Were these all other patients screened on admission to the HDU or were these patients who were positive on at least one body site and screened on admission to HDU?
Line 245 and 248 and elsewhere: Missing a comma before “etc.”
Table 1: Please include the full N of both groups in the table.
Table 1: It would be helpful to know if the treatment with antifungals was before C. auris was identified or that in response to C. auris.
Table 1: It is unclear if ‘two antifungal drugs’ means concurrent use or that one drug was tried and did not work so another was used. If two antifungals of the same drug class were used, is that counted as one or two antifungal drugs?
Table 1: Asymptomatic colonization is not typically treated. Does this mean that over two thirds of the colonized patients were determined to have infections caused by C. auris?
Table 1: The authors should clarify up front what kind of mortality is being presented (e.g., 30 day, all cause, in-hospital, etc).
Lines 276-277: More context around the environmental screening would be helpful. How many samples were collected? Were the same or similar samples taken from each room? Was sampling attempted from the environment of all patients with C. auris, of patients with clinical cases, of rooms where transmission was suspected, etc.? Did the authors find any patterns in which samples turned out to be positive (e.g. samples from one particular unit or samples from rooms of patients with multiple body sites positive, etc.)? When was cleaning performed compared to when the samples were taken? Ws the goal of environmental sampling to look for the relationship between skin colonization and environmental contamination or was it performed as a tool to evaluate how well the facility was cleaning?
Line 288-289: The outlier case is interesting and having a little more information about the epidemiology of it would be useful for interpretation. Had they received care elsewhere prior to getting care at this facility? Lines 368-370 state that the authors do not know the exact hospital they were transferred from but even knowing whether or not this patient was transferred from a hospital or came from home and whether they were positive on admission or found to be positive later in their hospitalization would be helpful.
Table 2: The text (line 138-139) says interpretations were based on EUCAST, but EUCAST is a 24 hour read due to known sharp increases in MICs at 48 hours that do not align with patient outcomes. More explanation is needed if the authors intend to use SIR interpretations comparing 24 to 48 hours since that is going outside of the standards set by CLSI and EUCAST.
Line 331-332: It is not clear if the treatment before, after, or in-between the two urine isolates for each of these patients. (That is, did resistance develop on treatment?)
Line 341-342: The effects of infection control changes as a result of COVID-19 on MDRO transmission is a hot topic right now and may be worth adding a sentence on whether these two events were likely tied together. Did the increase in cases in February 2020 coincide with changes in the facility’s infection control practices due to COVID-19? Perhaps bringing up in the conclusion that surveillance is being limited by COVID-19 would help shed light on the issue.
Line 354-355: I would not say the temperature probe outbreak suggest that axilla is universally colonized earlier than other body sites. Such a statement requires further examination and does not seem applicable to this investigation.
Line 355-358: Here and above, in order to interpret these percentages by body site, it would be helpful to see how many patients were sampled and how many were positive by each body site (perhaps in a table). The text makes it sound like patients were not all sampled in the same body sites, so the reader cannot tell what the denominators are for each site.
Line 405: Change anidulafunin to anidulafungin.
Author Response
Re: Manuscript ID: jof-987732-Alfouzan et al. 2020
Ms. Lily Guo
Editorial Office
Journal of Fungi
Dear Ms. Guo
Thank you very much for sending us the comments of the three reviewers on our above referenced manuscript. The manuscript has now been revised in light of the comments and suggestions of the three reviewers and all the changes made in the manuscript are highlighted in red font. We will like to add here that Table 3 was deleted from the manuscript during formatting at Journal of Fungi Editorial Office. Consequently, Table 2 has been replaced by the revised Table 2 while the revised Table 3 has been added in the revised manuscript. A point-by-point response to the various comments/suggestions of the Reviewer 1 is as follows.
Reviewer Comments:
Reviewer no. 1
Reviewer comments:
This is an overview of an investigation of an ongoing C. auris outbreak in an acute care hospital in Kuwait. The authors provide a well-rounded description of the patients and human and environmental samples from epidemiologic and laboratory perspectives. Their comprehensive approach to the investigation and subsequent description touches on several issues of interest in the field, including body site positivity, candidemia v. colonization or non-invasive infections, environmental contamination, patient outcomes, drug resistance, genetic relatedness, and, however briefly, how COVID-19 is affecting C. auris response.
Authors response: We thank the reviewer for the positive comments. No specific comments to respond to.
Reviewer comments:
Line 51-52: The authors should clarify this part of the sentence: “…its capability to evade immune response and easy person-to-person transmission through direct/indirect contact…” As it is, it reads to be like it is difficult to transmit and also that that paradoxically makes it a “formidable foe.” I think perhaps the intent is more that it is very easily transmitted by medical equipment and provider hands and thus still transmits easily despite the patients not being in direct contact with each other.
Authors response: The text has been modified for greater clarity, as suggested by the reviewer.
Reviewer comments:
Line 59: I am not sure “originating” is appropriate here as it might be subject to finder’s bias. We do not necessarily know that those clades originated in those regions.
Authors response: The text has been changed, as suggested by the reviewer.
Reviewer comments:
Line 93-94: This is suggesting that there were no cases that where asymptomatic and the authors happened to find C. auris rather than collecting samples to specifically look for this. However, this is common, so the authors may want to clarify whether this did not happen or whether such patients were counted as having clinical or colonized cases.
Line 93-94: There was a timeframe provided for the clinical cases, but not for the colonized cases. This should be added if available. For instance, are the authors only counting the point from which all patients were screened onward or did they begin screening select patients prior to that?
Authors response: The text has been modified for greater clarity, as suggested by the reviewer.
Reviewer comments:
Line 97: “Contacts” is defined here, but not used elsewhere in this paragraph to understand what happened with contacts. If they were saying that contacts were screened, I would recommend being more explicit or using the term “contact” elsewhere in that paragraph.
Authors response: The text has been modified, as suggested by the reviewer.
Reviewer comments:
Line 102: This line says environmental samples were obtained from “rooms/facilities,” but I thought that all sampling in this study was from the same facility. Should this be buildings or units instead of facilities?
Lines 103-105: Check plurality and punctuation in this sentence.
Authors response: The text has been modified, as suggested by the reviewer.
Reviewer comments:
Line 139: The citation here is 47, but the EUCAST citation in the references is 48. The authors must look through this document and make sure all citations are correctly lining up with their intended references. It looks like this is perhaps caused by the title “References” showing up at #1 in the reference list.
Authors response: We thank the Reviewer for pointing this out to us. Indeed, the text formatting done by the journal numbered ‘References” as #1 which had increased the number of all citations by 1 under References. This has been corrected.
Reviewer comments:
Line 210: Change isolated to isolates.
Authors response: The error has been corrected, as suggested by the reviewer.
Reviewer comments:
Figure 1: It is not clear how colonized cases were identified prior to the identification of the outbreak (i.e. prior to September 2018). Did those patients have unusual circumstances that caused the authors to want to screen them even though they were unaware of C. auris within the facility (e.g. healthcare exposure elsewhere in the country or abroad known to have C. auris)? This makes me continue to question the clinical v. colonized case definition, as asked about in my Line 93-94 question above.
Authors response: The colonized cases, before the outbreak was recognized, were detected during culture of different body fluids for the isolation of bacterial/fungal pathogens as part of routine patient care of hospitalized patients and were retrospectively included among colonized cases. This information has been included in response to the comments concerning lines 93-94 shown above.
Reviewer comments:
Figure 1: It might help to have more of a standard format for the epi curve. Here, it looks like there is a break between each month where no cases were found, but I am guessing it was more continuous than an oscillating up and down curve and these breaks are rather just a result of how to graphic is made.
Figure 1: It would be helpful to clarify if any patients who were first found to have C. auris due to a screening but then they subsequently developed C. auris candidemia. And, if so, how are these categorized? I’m wondering if that distinction would change the timeline for these cases at all.
Authors response: The figure legend has been expanded to include information, as requested by the reviewer.
Reviewer comments:
Line 227: Please check to see if IQR was meant instead of 95% CI in the parentheses.
Authors response: The text has been simplified, as suggested by the reviewer.
Reviewer comments:
Line 234-237: I would recommend putting the percentages next to the body site given the list is so long and indicating what the denominators were. Also, it is not clear if the authors are saying that the patients with candidemia were screened after they were known to have candidemia already or if patients who were screened and then went on to have candidemia. From 358-359 in the discussion makes it sound like the former, but this would be helpful to clarify earlier on. Also, were the patients with candidemia not screened at the other body sites or were they negative? Who were the “44 other patients”? Were these all other patients screened on admission to the HDU or were these patients who were positive on at least one body site and screened on admission to HDU?
Authors response: The number of subjects positive and the total number of subjects screened have been added. The patients with candidemia were also screened and those found positive at other sites are described in the text. The “44 other patients” were colonized subjects that were actively screened from December 2018 and this information has been added more explicitly, as suggested by the reviewer.
Reviewer comments:
Line 245 and 248 and elsewhere: Missing a comma before “etc.”
Authors response: The text has been modified, as suggested by the reviewer.
Reviewer comments:
Table 1: Please include the full N of both groups in the table.
Authors response: The number of cases in both groups have been added, as suggested by the reviewer.
Reviewer comments:
Table 1: It would be helpful to know if the treatment with antifungals was before C. auris was identified or that in response to C. auris.
Authors response: The information has been provided, as suggested by the reviewer.
Reviewer comments:
Table 1: It is unclear if ‘two antifungal drugs’ means concurrent use or that one drug was tried and did not work so another was used. If two antifungals of the same drug class were used, is that counted as one or two antifungal drugs?
Authors response: The text has been modified for greater clarity, as requested by the reviewer.
Reviewer comments:
Table 1: Asymptomatic colonization is not typically treated. Does this mean that over two thirds of the colonized patients were determined to have infections caused by C. auris?
Authors response: The colonized patients were treated empirically due to their critical condition and this information has been added in the revised manuscript, as requested by the reviewer.
Reviewer comments:
Table 1: The authors should clarify up front what kind of mortality is being presented (e.g., 30 day, all cause, in-hospital, etc).
Authors response: The information has been provided, as suggested by the reviewer.
Reviewer comments:
Lines 276-277: More context around the environmental screening would be helpful. How many samples were collected? Were the same or similar samples taken from each room? Was sampling attempted from the environment of all patients with C. auris, of patients with clinical cases, of rooms where transmission was suspected, etc.? Did the authors find any patterns in which samples turned out to be positive (e.g. samples from one particular unit or samples from rooms of patients with multiple body sites positive, etc.)? When was cleaning performed compared to when the samples were taken? Ws the goal of environmental sampling to look for the relationship between skin colonization and environmental contamination or was it performed as a tool to evaluate how well the facility was cleaning?
Authors response: More information has been provided, as suggested by the reviewer.
Reviewer comments:
Line 288-289: The outlier case is interesting and having a little more information about the epidemiology of it would be useful for interpretation. Had they received care elsewhere prior to getting care at this facility? Lines 368-370 state that the authors do not know the exact hospital they were transferred from but even knowing whether or not this patient was transferred from a hospital or came from home and whether they were positive on admission or found to be positive later in their hospitalization would be helpful.
Authors response: We have provided additional information retrieved from the hospital records for this patient in the ‘Discussion’ section, as suggested by the reviewer.
Reviewer comments:
Table 2: The text (line 138-139) says interpretations were based on EUCAST, but EUCAST is a 24 hour read due to known sharp increases in MICs at 48 hours that do not align with patient outcomes. More explanation is needed if the authors intend to use SIR interpretations comparing 24 to 48 hours since that is going outside of the standards set by CLSI and EUCAST.
Authors response: We agree with the reviewer. Consequently, the 48 h data have been deleted and both Table 2 and Table 3 have been reorganized in the revised manuscript.
Reviewer comments:
Line 331-332: It is not clear if the treatment before, after, or in-between the two urine isolates for each of these patients. (That is, did resistance develop on treatment?)
Authors response: The information has been provided, as requested by the reviewer.
Reviewer comments:
Line 341-342: The effects of infection control changes as a result of COVID-19 on MDRO transmission is a hot topic right now and may be worth adding a sentence on whether these two events were likely tied together. Did the increase in cases in February 2020 coincide with changes in the facility’s infection control practices due to COVID-19? Perhaps bringing up in the conclusion that surveillance is being limited by COVID-19 would help shed light on the issue.
Authors response: The information has been provided, as requested by the reviewer.
Reviewer comments:
Line 354-355: I would not say the temperature probe outbreak suggest that axilla is universally colonized earlier than other body sites. Such a statement requires further examination and does not seem applicable to this investigation.
Authors response: The text has been modified, as suggested by the reviewer.
Reviewer comments:
Line 355-358: Here and above, in order to interpret these percentages by body site, it would be helpful to see how many patients were sampled and how many were positive by each body site (perhaps in a table). The text makes it sound like patients were not all sampled in the same body sites, so the reader cannot tell what the denominators are for each site.
Authors response: The number of subjects positive and the total number of subjects screened have been added. The denominator was the 44 subjects that were actively screened from December 2018 and this information has been added more explicitly, as suggested by the reviewer.
Reviewer comments:
Line 405: Change anidulafunin to anidulafungin.
Authors response: The mistake has been corrected.
The manuscript is much improved and we thank the three reviewers for this improvement. We hope that it will also meet the approval of the three reviewers and the manuscript handling Editor.
Looking forward to hearing further from you
With kind regards
Dr. Wadha Alfouzan
Associate Professor,
Department of Microbiology
Faculty of Medicine, Kuwait University
Reviewer 2 Report
Alfouzan et al., present a very nice study detailing an outbreak of C. auris in a secondary-care unit in Kuwait.
My main critisim of the introduction would be that many of the sentences are too long spanning 5-8 lines.
The laboratory methods should be expanded not just continuously referring to previously published methods. The companies reagents were purchased from especially the azoles should be mentioned. Images of the breakpoints and BMD plates would be advantageous in driving home the points of different levels of drug resistance. In addition a schematic of ERG11 and FKS1 showing the regions of interest for point mutations/deletions would better guide the reader.
Section 2.3 lines 176-191 is this necessary as it is summarised in Table 1.
Much more detail on the statisitial analysis is required.
Figure 1 could be improved to a more detailed schema rather than an arrow
Figure 2 is out of focus
The references need to be checked - number 1 is:reference therefore the numbers are off throughout the main body of the text.
Author Response
Re: Manuscript ID: jof-987732-Alfouzan et al. 2020
Ms. Lily Guo
Editorial Office
Journal of Fungi
Dear Ms. Guo
Thank you very much for sending us the comments of the three reviewers on our above referenced manuscript. The manuscript has now been revised in light of the comments and suggestions of the three reviewers and all the changes made in the manuscript are highlighted in red font. We will like to add here that Table 3 was deleted from the manuscript during formatting at Journal of Fungi Editorial Office. Consequently, Table 2 has been replaced by the revised Table 2 while the revised Table 3 has been added in the revised manuscript. A point-by-point response to the various comments/suggestions of the Reviewer 2 is as follows.
Reviewer no. 2
Reviewer comments:
My main critisim of the introduction would be that many of the sentences are too long spanning 5-8 lines.
Authors response: The larger sentences have been shortened, as suggested by the reviewer.
Reviewer comments:
The laboratory methods should be expanded not just continuously referring to previously published methods. The companies reagents were purchased from especially the azoles should be mentioned. Images of the breakpoints and BMD plates would be advantageous in driving home the points of different levels of drug resistance. In addition a schematic of ERG11 and FKS1 showing the regions of interest for point mutations/deletions would better guide the reader.
Authors response: The laboratory methods have been expanded and more details have been provided, as suggested by the reviewer. The antifungal susceptibility testing was carried out by using commercial kit obtained from Merlin Diagnostica GmbH, Bornheim, Germany which contained all the antifungal drugs tested and the supplier name has been provided in the text. The mutations detected in ERG11 and FKS1 are well-known and since only few isolates had FKS1 mutations, we do not feel that their schematic representation is warranted in this manuscript.
Reviewer comments:
Section 2.3 lines 176-191 is this necessary as it is summarised in Table 1.
Authors response: This portion describes the infection control measures adopted during the outbreak and these details are not present in Table 1.
Reviewer comments:
Much more detail on the statisitial analysis is required.
Authors response: More details have been provided and the statistical program used to find significant differences for the variables among different groups is described. as suggested by the reviewer. In any case, there were no major differences in the two groups, so no significantly different findings were noted among the candidemia and colonized patients in this study.
Reviewer comments:
Figure 1 could be improved to a more detailed schema rather than an arrow
Authors response: Figure 1 only describes the tine-line for the identification of candidemia and colonized patients during the outbreak and its legend has been improved, as also requested by Reviewer no. 1
Reviewer comments:
Figure 2 is out of focus
Authors response: The original figure with required dpi has been provided.
Reviewer comments:
The references need to be checked - number 1 is:reference therefore the numbers are off throughout the main body of the text.
Authors response: We thank the Reviewer for pointing this out to us. Indeed, the text formatting done by the journal numbered ‘References” as #1 which had increased the number of all citations by 1 under References. This has been corrected.
The manuscript is much improved and we thank the three reviewers for this improvement. We hope that it will also meet the approval of the three reviewers and the manuscript handling Editor.
Looking forward to hearing further from you
With kind regards
Dr. Wadha Alfouzan
Associate Professor,
Department of Microbiology
Faculty of Medicine, Kuwait University
Reviewer 3 Report
Alfouzan and colleagues described here the Candida auris outbreak in a major secondary-care hospital in Kuwait. There is fact fact fungal infections, and particularly opportunistic fungal infections, such as Candida spa infections, have triggered an increasingly attention by both medical and scientists community, given the huge impact they have posed for human health.
In this case, and despite Candida albicans have been the main focus of studies, given its associated virulence, the fact is that an increasingly amount of non-albicans Candida species have revealed to impact human health at such or even higher level. Here, the authors described the molecular epidemiology of a C. auris outbreak over 18 months. In general, the work is interesting and suitable for publication in this journal. There are however some aspects that still need to be addressed:
- l. 31-33: please rewrite this sentence, is incomplete
- methods: the authors referred that in this report 71 patients (17 candemic and 54 colonized) were included. However, no mention was done to inclusion and exclusion criteria (i.e. were such patients under effect of immunosuppressors, etc.). In addition, was this study approved by the ethics committee? please provide the study approval number. Moreover, was informed consent obtained from such patients?
- l. 195-196: Why groups characterization is presented in statistical analysis subsection and not on sample collection subsection? in addition, what was the rationale for defining the 4 groups of comorbidities? Please explain.
- results: how was the required sample size calculated to ensure statistical significance of the data obtained?
- Table 1: data obtained between both groups (candemic vs colonized) were adjusted for underlying disease? mortality rate was also adjusted for the diseases that patients present, given that most of them did not die from the infection but other causes?
- Table 1: regarding the "treatment with antifungals", what type of combinations were more frequent? and what were the most common antifungals prescribed in such patients? did they present side effects as a result of drug administration?
- l. 328-331: how was this finding obtained?
- l. 336-342: was this protocol previously applied and/or developed in other places?
- l. 357: "among 44 patients", are authors referring to candemic or colonized patients? or both? please explain
- what are the main conclusions and upcoming perspective to be highlighted based on data obtained here?
Author Response
Re: Manuscript ID: jof-987732-Alfouzan et al. 2020
Ms. Lily Guo
Editorial Office
Journal of Fungi
Dear Ms. Guo
Thank you very much for sending us the comments of the three reviewers on our above referenced manuscript. The manuscript has now been revised in light of the comments and suggestions of the three reviewers and all the changes made in the manuscript are highlighted in red font. We will like to add here that Table 3 was deleted from the manuscript during formatting at Journal of Fungi Editorial Office. Consequently, Table 2 has been replaced by the revised Table 2 while the revised Table 3 has been added in the revised manuscript. A point-by-point response to the various comments/suggestions of the Reviewer 3 is as follows.
Reviewer no. 3
Reviewer comments:
Alfouzan and colleagues described here the Candida auris outbreak in a major secondary-care hospital in Kuwait. There is fact fact fungal infections, and particularly opportunistic fungal infections, such as Candida spa infections, have triggered an increasingly attention by both medical and scientists community, given the huge impact they have posed for human health.
In this case, and despite Candida albicans have been the main focus of studies, given its associated virulence, the fact is that an increasingly amount of non-albicans Candida species have revealed to impact human health at such or even higher level. Here, the authors described the molecular epidemiology of a C. auris outbreak over 18 months. In general, the work is interesting and suitable for publication in this journal.
Authors response: We thank the reviewer for the positive comments. No specific comments to respond to.
There are however some aspects that still need to be addressed:
Reviewer comments:
- l. 31-33: please rewrite this sentence, is incomplete
- Reviewer comments:
- Authors response: The text has been modified, as suggested by the reviewer.
- methods: the authors referred that in this report 71 patients (17 candemic and 54 colonized) were included. However, no mention was done to inclusion and exclusion criteria (i.e. were such patients under effect of immunosuppressors, etc.). In addition, was this study approved by the ethics committee? please provide the study approval number. Moreover, was informed consent obtained from such patients?
- Reviewer comments:
- Authors response: All patients that yielded C. auris from bloodstream or other anatomic sites during January 2018 and June 2019 were included in the study. This information has been added in the revised manuscript, as suggested by the reviewer. Informed verbal consent was obtained from hospitalized patients before collection of clinical specimens including blood as part of routine patient care and diagnostic work-up for the isolation and antifungal susceptibility testing of fungal pathogens. Since the results are reported in this manuscript on deidentified samples without revealing patient identity, specific ethical approval for this study was not needed.
- l. 195-196: Why groups characterization is presented in statistical analysis subsection and not on sample collection subsection? in addition, what was the rationale for defining the 4 groups of comorbidities? Please explain.
- Reviewer comments:
- Authors response: The group characterization has been moved to the sample collection subsection as suggested by the reviewer. The comorbidities were defined into four major groups as they were more frequently seen among both clinical cases and colonized patients.
- results: how was the required sample size calculated to ensure statistical significance of the data obtained?
- Reviewer comments:
- Authors response: All patients that yielded C. auris from bloodstream or other anatomic sites during January 2018 and June 2019 were included in the study. This information has been added in the revised manuscript.
- Table 1: data obtained between both groups (candemic vs colonized) were adjusted for underlying disease? mortality rate was also adjusted for the diseases that patients present, given that most of them did not die from the infection but other causes?
- Authors response: The mortality rate reported in the study is the crude, in-hospital mortality and was not adjusted for other underlying diseases present in both candidemia and colonized subjects. Reviewer comments:Authors response: Caspofungin was the most common antifungal drug prescribed and only two candidemia patients received treatment with more than one class of antifungal drug and this information has been provided in the manuscript. Some patients did have side effects of drug administration, however, this information is beyond the scope of this manuscript.Reviewer comments:
- Table 1: regarding the "treatment with antifungals", what type of combinations were more frequent? and what were the most common antifungals prescribed in such patients? did they present side effects as a result of drug administration?
- l. 328-331: how was this finding obtained?
- Reviewer comments:
- Authors response: The information was obtained from the date of isolation of various urine isolates and the period of treatment with echinocandins.
- l. 336-342: was this protocol previously applied and/or developed in other places?
- Reviewer comments:
- Authors response: The infection prevention and control measures were instituted according to the U.S. Centers of Disease Control and Prevention (CDC) guidelines, and Kuwait Infection Control Directorate (Ministry of Health) as reported under ‘Materials and Methods’ along with their cited references.
- l. 357: "among 44 patients", are authors referring to candemic or colonized patients? or both? please explain
- Reviewer comments:
- Authors response: These were colonized patients. This information has been added in the revised manuscript, as requested by the reviewer.
- what are the main conclusions and upcoming perspective to be highlighted based on data obtained here? Associate Professor, Faculty of Medicine, Kuwait University
- Department of Microbiology
- Dr. Wadha Alfouzan
- With kind regards
- Looking forward to hearing further from you
- The manuscript is much improved and we thank the three reviewers for this improvement. We hope that it will also meet the approval of the three reviewers and the manuscript handling Editor.
- Authors response: The data presented on a C. auris outbreak have shown that only a minority (17 of 71, 23.9) of patients developed candidemia. Also, only urine isolates from two patients, treated with caspofungin and/or anidulafungin, developed echinocandin resistance with concomitant FKS1 mutations. Despite treatment, an overall crude mortality rate of ~50% was seen among both candidemia and colonized patients. Our inability to decolonize patients due to unique properties of C. auris to cause prolonged host colonization contributed to the sustenance of the organism causing invasive infections at locations other than the HDU. Despite continuous surveillance and enforcement of infection control measures, new cases continued to occur, challenging the containment efforts. These aspects are highlighted at the end of the manuscript under the concluding paragraph.
Round 2
Reviewer 3 Report
All comments raised were addressed